# Metabolic and pathologic profiles of human LSS deficiency recapitulated in mice

**Yoichi Wada**[1], **Atsuo Kikuchi**[1]*, **Akimune Kaga**[2], **Naoki Shimizu**[3], **Junya Ito**[3], **Ryo Onuma**[3], **Fumiyoshi Fujishima**[4], **Eriko Totsune**[1], **Ryo Sato**[1], **Tetsuya Niihori**[5], **Matsuyuki Shirota**[6], **Ryo Funayama**[7], **Kota Sato**[8,9], **Toru Nakazawa**[8,9,10,11,12], **Keiko Nakayama**[7], **Yoko Aoki**[5], **Setsuya Aiba**[13], **Kiyotaka Nakagawa**[3], **Shigeo Kure**[1]

1 Department of Pediatrics, Tohoku University Graduate School of Medicine, Sendai, Miyagi, Japan, 2 Department of Pediatrics, Tohoku Kosai Hospital, Sendai, Miyagi, Japan, 3 Food and Biodynamic Chemistry Laboratory, Graduate School of Agricultural Science, Tohoku University, Sendai, Miyagi, Japan, 4 Department of Anatomic Pathology, Tohoku University Graduate School of Medicine, Sendai, Miyagi, Japan, 5 Department of Medical Genetics, Tohoku University Graduate School of Medicine, Sendai, Miyagi, Japan, 6 Division of Interdisciplinary Medical Sciences, United Centers for Advanced Research and Translational Medicine, Tohoku University Graduate School of Medicine, Sendai, Miyagi, Japan, 7 Division of Cell Proliferation, United Centers for Advanced Research and Translational Medicine, Tohoku University Graduate School of Medicine, Sendai, Miyagi, Japan, 8 Department of Ophthalmology, Tohoku University Graduate School of Medicine, Sendai, Miyagi, Japan, 9 Collaborative Program for Ophthalmic Drug Discovery, Tohoku University Graduate School of Medicine, Sendai, Miyagi, Japan, 10 Department of Retinal Disease Control, Tohoku University Graduate School of Medicine, Sendai, Miyagi, Japan, 11 Department of Advanced Ophthalmic Medicine, Tohoku University Graduate School of Medicine, Sendai, Miyagi, Japan, 12 Department of Ophthalmic Imaging and Information Analytics, Tohoku University Graduate School of Medicine, Sendai, Miyagi, Japan, 13 Department of Dermatology, Tohoku University Graduate School of Medicine, Sendai, Miyagi, Japan

* akikuchi-thk@umin.ac.jp

**Data Availability Statement:** All relevant data are within the manuscript and its Supporting Information files.

## Abstract

Skin lesions, cataracts, and congenital anomalies have been frequently associated with inherited deficiencies in enzymes that synthesize cholesterol. Lanosterol synthase (LSS) converts (S)-2,3-epoxysqualene to lanosterol in the cholesterol biosynthesis pathway. Biallelic mutations in *LSS* have been reported in families with congenital cataracts and, very recently, have been reported in cases of hypotrichosis. However, it remains to be clarified whether these phenotypes are caused by LSS enzymatic deficiencies in each tissue, and disruption of LSS enzymatic activity *in vivo* has not yet been validated. We identified two patients with novel biallelic *LSS* mutations who exhibited congenital hypotrichosis and midline anomalies but did not have cataracts. We showed that the blockade of the LSS enzyme reaction occurred in the patients by measuring the (S)-2,3-epoxysqualene/lanosterol ratio in the forehead sebum, which would be a good biomarker for the diagnosis of LSS deficiency. Epidermis-specific *Lss* knockout mice showed neonatal lethality due to dehydration, indicating that LSS could be involved in skin barrier integrity. Tamoxifen-induced knockout of *Lss* in the epidermis caused hypotrichosis in adult mice. Lens-specific *Lss* knockout mice had cataracts. These results confirmed that LSS deficiency causes hypotrichosis and cataracts due to loss-of-function mutations in *LSS* in each tissue. These mouse models will lead to the elucidation of the pathophysiological mechanisms associated with disrupted LSS and to the development of therapeutic treatments for LSS deficiency.

**Funding:** This research was supported by the Japan Agency for Medical Research and Development (AMED) through grant number JP17ek0109151 (Initiative on Rare and Undiagnosed Diseases [IRUD] to S.K.) and JSPS KAKENHI grant number JP17K10044 (to At.K. and K.N.). The funders had no role in study design, data collection and analysis, decision to publish, or preparation of the manuscript.

**Competing interests:** The authors have declared that no competing interests exist.

## Author summary

Skin lesions, cataracts, and congenital anomalies have been frequently associated with inherited deficiencies of cholesterol synthetic enzymes. *LSS* encodes lanosterol synthase, an enzyme in the cholesterol biosynthesis pathway, and biallelic mutations in *LSS* have been reported in families with congenital cataracts and hypotrichosis. However, it remains to be clarified whether these phenotypes are caused by LSS enzymatic deficiency in each tissue, and disruption of LSS enzymatic activity *in vivo* has not yet been validated. We showed that LSS metabolic inhibition in patients with biallelic *LSS* mutations and congenital hypotrichosis *in vivo* by measuring metabolites of the LSS enzyme in the forehead sebum, which would be good biomarkers for the diagnosis of LSS deficiency. We recapitulated hypotrichosis and cataracts by creating tissue-specific *Lss* knockout mice. Our mouse studies confirmed that LSS deficiency causes hypotrichosis and cataracts due to loss-of-function mutations in *LSS* in each tissue. These mice will lead to the elucidation of the pathophysiological mechanisms associated with disrupted LSS and to the development of therapeutic treatments for LSS deficiency.

## Introduction

Cholesterol is essential for the regulation of biological functions in animals. In cell membranes, cholesterol enhances fluidity, reduces permeability and is a constituent of lipid rafts [1]. Moreover, cholesterol or its intermediates are precursors of vitamin D, steroid hormones, and bile acids. Cholesterol itself alters the activity of proteins such as sonic hedgehog (Shh) [2][3] which is involved in embryogenesis and tissue growth. In the cholesterol synthesis pathway, beginning with acetyl-CoA, lanosterol synthase (LSS) converts (S)-2,3-epoxysqualene to lanosterol [4] (S1 Fig). Biallelic mutations in *LSS* were first reported in families with congenital cataracts [5,6]. In addition, very recently, hypotrichosis has been reported to be an associated phenotype with biallelic *LSS* mutations [7,8]. Out of 24 patients, all had either cataracts (5/24) or hypotrichosis (20/24), and one had both [5–8]. Thus, cataracts and hypotrichosis are core symptoms in patients with biallelic *LSS* mutations. However, the pathophysiology in patients largely remains to be clarified. First, it is not proven that biallelic *LSS* mutations cause LSS deficiency because the metabolite profiles in the previously reported patients have not been compatible with LSS enzymatic dysfunctions. The cholesterol or the intermediate levels in serum or plasma did not significantly change between the patients and controls [7,8]. Second, animal models that reproduce the LSS-deficient state remain to be developed. Although the Shumiya cataract rat has biallelic *Lss* mutations [9], this rat has additional mutations in *Fdft1*, which encodes an upstream enzyme in the cholesterol synthetic pathway. Moreover, the metabolic profiles of the lens were inconsistent with LSS enzymatic inhibition in the Shumiya cataract rat. There is no report of model mice that reproduce cataract or hypotrichosis due to the embryonic lethality of the *Lss* null mice [10]. Collectively, it has not been established whether biallelic *LSS* mutations cause LSS enzymatic deficiency and the local symptoms in each tissue or if they lead to systemic effects on abnormal metabolites, and this is due to the lack of an appropriate model animal and the difficulty in measuring metabolites of the LSS enzyme.

Here, we report two siblings with novel biallelic *LSS* mutations who presented congenital hypotrichosis, midline anomalies, such as cleft palate and agenesis of the corpus callosum, and no cataracts. The aims of this study were to confirm an LSS enzymatic disruption, that is, the

presence of LSS deficiency, in patients with biallelic *LSS* mutation and to create *Lss*-deficient mice to elucidate whether LSS enzymatic depletion in each tissue is responsible for the symptoms in that tissue.

## Results

### Clinical information of patients with congenital hypotrichosis and anomalies

We studied two sibling patients with congenital hypotrichosis and several anomalies (Fig 1a). The parents and the first child had no medical problems, including congenital anomalies. The first patient (patient II-2) was the second boy in a nonconsanguineous Japanese family (Fig 1b). This patient was delivered at 36 weeks of gestation due to low amniotic fluid and fetal growth restriction (FGR). His birth weight was 1,848 g (-2.26 SD), and height was 42.0 cm (-2.03 SD). At birth, mild ichthyosis was observed on his chest, abdomen, back, and upper extremities (S2 Fig). This ichthyosis disappeared by the age of 2 months. His body hair never grew despite treatment, such as topical vitamin D and steroid therapy. He had intermittent exotropia, hypospadias and partial agenesis of the corpus callosum (S3 Fig). At the most recent follow-up (18 years old), his height was 157.1 cm (-2.35 SD), and his body weight was 66.1 kg (0.54 SD), showing significant growth retardation.

The second patient (patient II-3) was the third boy in the family (Fig 1b). He was delivered at 32 weeks of gestation to FGR. His birth weight was 1,180 g (-2.86 SD), and height was 37.0 cm (-2.36 SD). As with the first patient, this patient exhibited treatment-resistant hypotrichosis and had a cleft palate and umbilical herniation. MR images of his head demonstrated a normal corpus callosum (S3 Fig). At the most recent follow-up (16 years old), his height was 147.8 cm (-3.69 SD), and his body weight was 71.0 kg (1.09 SD), showing growth retardation similar to patient II-2.

Their motor and mental development of the patients are normal, and they do not present with cataracts. Histological analysis showed hypoplastic hair follicles, psoriasiform acanthosis, spongiosis, and hyperplastic sebaceous glands (Fig 1c–1f). The hair follicles were hypoplastic, similar to what is observed with lanugo. Ultrastructural examination by transmission electron microscopy (TEM) revealed that the number of hair matrix cells or melanin granules was decreased, and that the inner root sheath was thin (S4 Fig). The basement layer cells tightly adhered to each other, and the desmosomes developed normally.

### *LSS* was identified as a single candidate gene by whole-exome sequencing

We performed family-based exome sequencing, for which we included the affected patients, an unaffected sibling and the unaffected parents. The mean depth of each exome was approximately 120. We identified only one candidate gene, *LSS* (NM_002340.5), for an autosomal recessive model in both patients (S1 Table). No rare *de novo* variants or rare X-linked variants shared by the patients were identified. Sanger sequencing validated compound heterozygous mutations in *LSS*, namely, c.[530G>A];[701_716del] (p.[Arg177Gln]; [Arg234Profs*2]), both of which were absent or extremely rare in ExAC, gnomAD, and the Human Genetic Variation Database (HGVD). The missense variant, p.Arg177Gln, was also predicted to be deleterious or pathogenic *in silico* (combined annotation-dependent depletion score, SIFT, Polyphen-2, and MutationTaster) (S2 Table), and it affects evolutionarily conserved residues (S5 Fig).

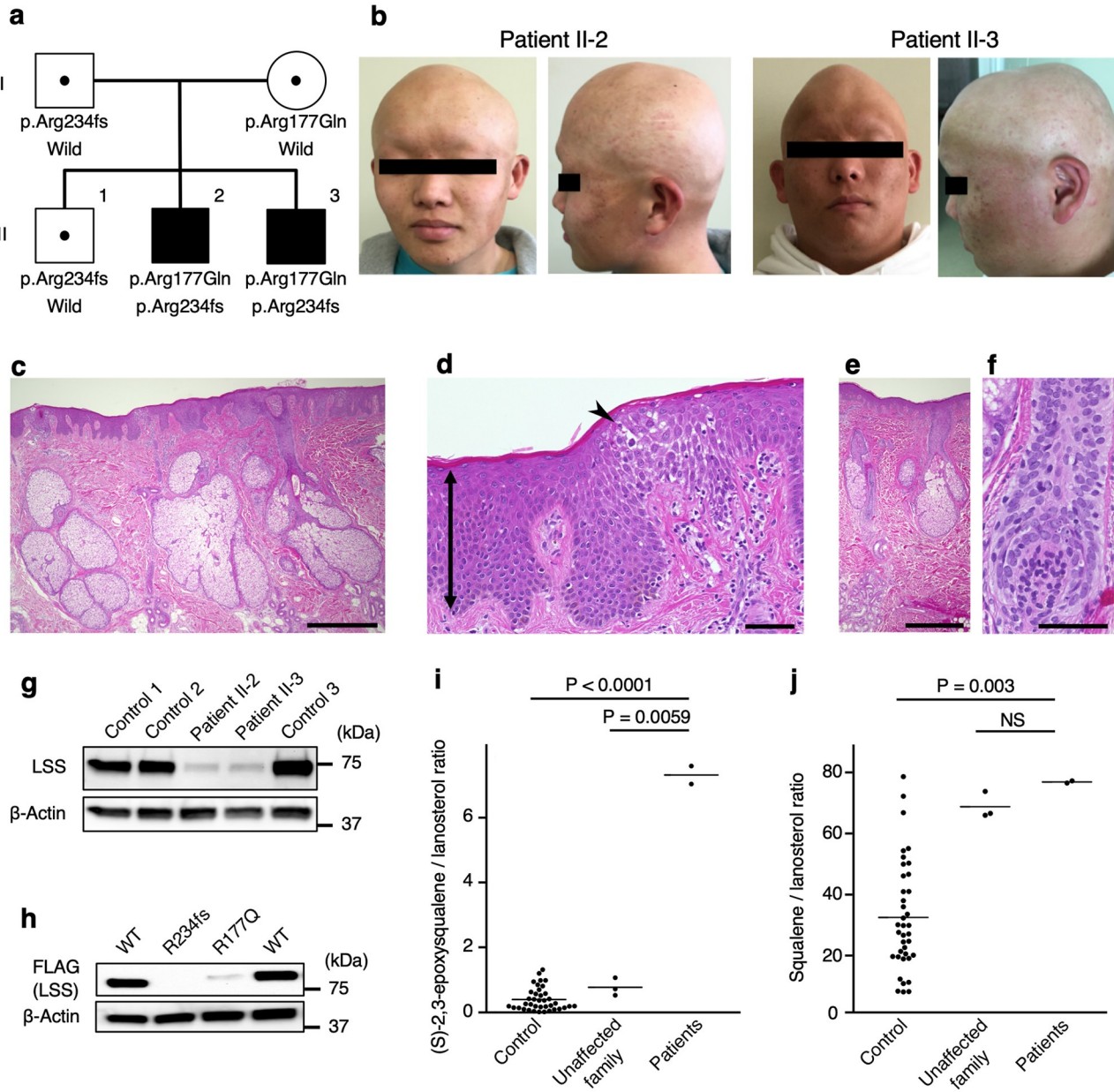

**Fig 1. Identification of biallelic *LSS* mutations in patients with hypotrichosis; the *LSS* mutations of the patients were hypomorphic and led to decreased enzymatic activity *in vivo*, and analysis of skin pathology revealed acanthosis and hypoplastic hair follicles. a**, Pedigree of the family. **b**, Photographs of patients II-2 and II-3. **c-f**, Histopathology of skin from patients II-2 and II-3. The epidermises exhibited psoriasiform acanthosis (arrow) and spongiosis (arrowhead). Scale bars = 600 μm (c, e), 60 μm (d, f). **g**, Immunoblot analysis of LSS expression in PBMCs from patients and controls. **h**, Overexpressed FLAG-LSS protein was detected with an anti-FLAG antibody. β-Actin was used as a loading control. Representative results from three independent experiments are shown. WT, wild type. **i, j**, LC-MS/MS analysis of (S)-2,3-epoxysqualene to lanosterol (i) and squalene to lanosterol (j) ratios in forehead sebum from the patients (n = 2), unaffected family members (n = 3), and controls (n = 37). P-values were calculated using Tukey-Kramer tests.

## *In vitro* and *in vivo* assays revealed that the *LSS* mutations were hypomorphic and led to decreased enzymatic activity

An immunoblot analysis showed decreased expression of LSS protein in peripheral blood mononuclear cells (PBMCs) in the patients compared to the controls (Fig 1g). The

c.701_716del (p.Arg234Profs*2) transcript was detected at a much lower level than the wild-type allele in the Epstein-Barr virus-transformed lymphoblastoid cell lines (EBV-LCLs) from the patients. Suppression of nonsense-mediated mRNA decay (NMD) by cycloheximide treatment increased the levels of the frameshifted transcript, indicating that the frameshifted allele resulted in an unstable transcript that was subjected to NMD (S6 Fig). To test the stability of the mutant LSS proteins, we overexpressed wild-type or mutant FLAG-tagged LSS in HeLa cells. Exogenous expression of both LSS mutants was markedly reduced (Fig 1h). The result of in vitro LSS expression suggested that the mutant protein stability was affected, although caution must be taken because overexpression studies sometimes do not reflect the protein stability of the mutants under physiological conditions. Next, we measured the levels of intermediate metabolites of cholesterol synthesis in human specimens to show LSS enzymatic inhibition *in vivo*. In plasma, we detected only (S)-2,3-epoxysqualene but not lanosterol by LC-MS/MS analysis (S7 Fig). By contrast, in sebum from the skin, namely, the affected organ in the patients, we successfully detected squalene, (S)-2,3-epoxysqualene, and lanosterol by LC-MS/MS analysis. The ratios of (S)-2,3-epoxysqualene to lanosterol, that is, the ratios of the substrate to the product of LSS catalysis, in the sebum from the affected patients were significantly elevated compared to those of the unaffected family members and healthy volunteers (Fig 1i). The ratios of the squalene to lanosterol were not different between the patients and the unaffected family members (Fig 1j). Cholesterol could not be detected in the sebum of all of them, and the serum levels were not significantly different among the groups (S3 Table).

## *Lss* epidermis-specific knockout mice were neonatal lethal because of skin barrier dysfunction that led to severe dehydration

To confirm that loss-of-function of the cutaneous LSS protein causes hypotrichosis, we created epidermis-specific *Lss* knockout mice; *Lss* constitutive knockout mice were previously reported to be embryonic lethal, as mentioned above. *Lss flox/flox;Keratin14-Cre/+* ($Lss^{f/f}$-*K14*) mice were obtained from breeding *Lss wild/flox;K14-Cre/+* male mice and *Lss flox/flox ($Lss^{f/f}$)* female mice (S8 Fig). The $Lss^{f/f}$-*K14* mice did not have macroscopic whiskers (Fig 2a). Genotype distribution of the $Lss^{f/f}$-*K14* mice and the litters at 18.5 days post coitum were observed to conform to a Mendelian distribution (based on 51 litters, S4 Table). Quantitative PCR (qPCR) analysis showed that *Lss* mRNA levels in the $Lss^{f/f}$-*K14* mice were significantly lower than those in the $Lss^{f/f}$ littermates in the epidermis but not significantly different in the liver (Fig 2b). Lipid measurements showed that (S)-2,3-epoxysqualene was present in the skin epidermis of the $Lss^{f/f}$-*K14* mice but not in the skin of the $Lss^{f/f}$ littermates (Fig 2c). Lanosterol and squalene were undetectable in the epidermis of both the $Lss^{f/f}$-*K14* and $Lss^{f/f}$ mice. Cholesterol was not significantly different between the epidermis of $Lss^{f/f}$-*K14* and $Lss^{f/f}$ mice (S9 Fig).

Their hair growth could not be investigated because all of the $Lss^{f/f}$-*K14* mice died within a few hours after birth. The body weight loss exhibited by these mice was significantly greater than that of $Lss^{wild/f}$-*K14* and $Lss^{f/f}$ mice, and it was greater than 10% until 6 hours after birth (Fig 2d); thus, severe dehydration immediately after birth was suspected as the cause of death. A dye penetration assay using toluidine blue showed that the $Lss^{f/f}$-*K14* mice exhibited higher levels of staining than the $Lss^{f/f}$ littermates (Fig 2e), which indicated outside-in skin barrier dysfunction in the $Lss^{f/f}$-*K14* mice. Transepidermal water loss (TEWL) was significantly higher in the $Lss^{f/f}$-*K14* mice than it was in the $Lss^{f/f}$ littermates, which indicated inside-out skin barrier impairment in the $Lss^{f/f}$-*K14* mice (Fig 2f). A light microscopic study did not show apparent differences between the $Lss^{f/f}$-*K14* and $Lss^{f/f}$ mice (S10a Fig). Expression of the LSS protein was decreased in the epidermal basement membrane of the $Lss^{f/f}$-*K14* mice compared to what was observed in the $Lss^{f/f}$ littermates by immunohistochemistry (S10b Fig). The number of

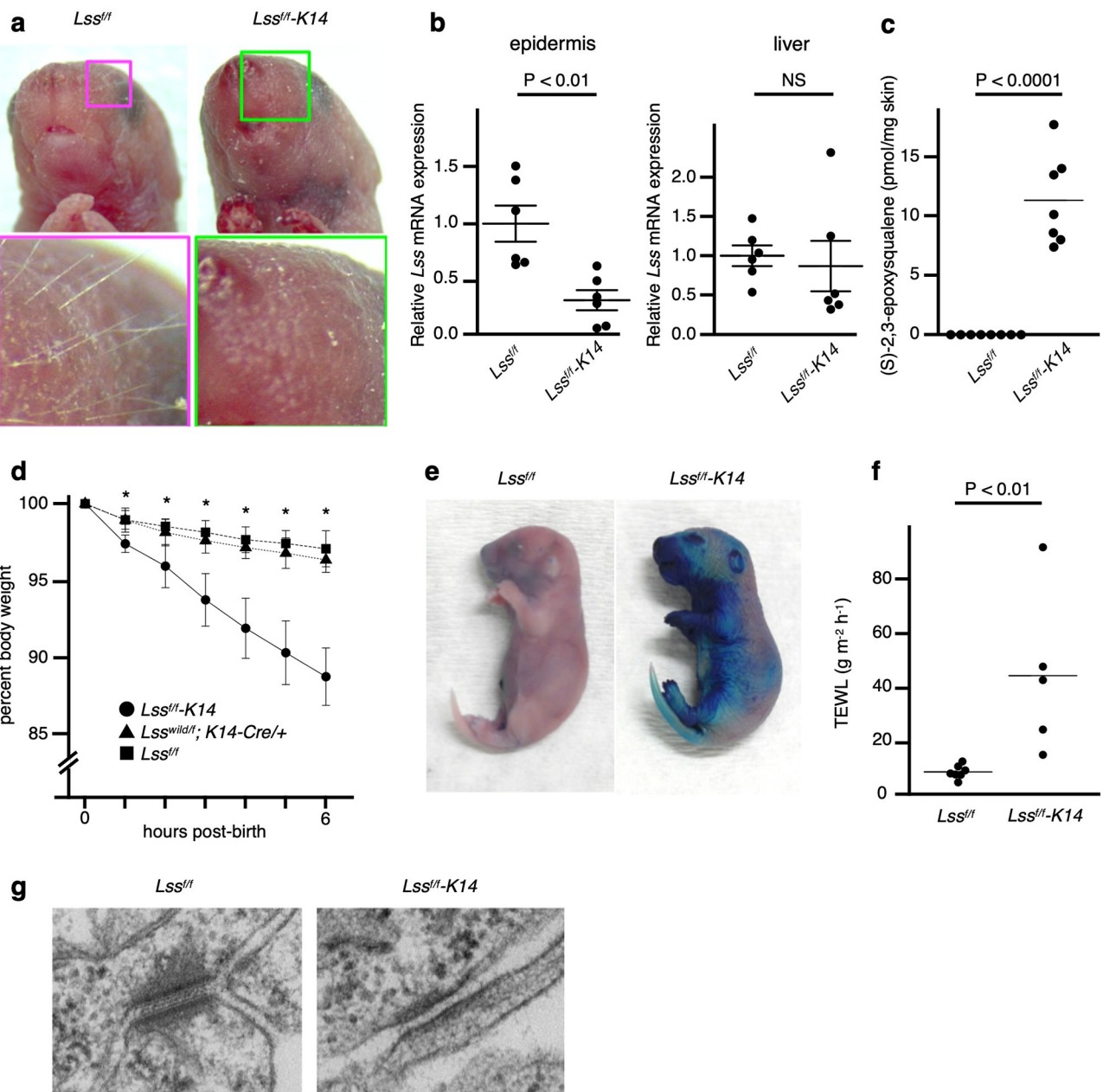

**Fig 2. Epidermis-specific *Lss* knockout neonatal mice exhibited no vibrissae and died of severe dehydration due to skin barrier dysfunction. a,** Facial appearance of a neonatal *Lss^{f/f}* or *Lss^{f/f}-K14* mouse. The magnified images show the presence of whiskers in an *Lss^{f/f}* mouse but not in an *Lss^{f/f}-K14* mouse. **b,** Quantitative PCR analysis of *Lss* in the epidermis (left) and liver (right) of newborn *Lss^{f/f}-K14* or *Lss^{f/f}* mice. Relative amounts of mRNA were determined using the comparative Ct method. Each bar represents the mean ± SEM (n = 6 per group). P-values were calculated using Student's *t*-tests. NS, not significant. **c,** Lipid analysis of newborn *Lss^{f/f}-K14* (n = 7) or *Lss^{f/f}* (n = 8) skin for (S)-2,3-epoxysqualene. All bars represent the means. P-values were calculated using Student's *t*-tests. **d,** Percent body weight based on the birth weights over time. The body weight loss of *Lss^{f/f}*-K14 mice (n = 5), which was over 10% until 6 hours after birth, was significantly higher than the loss observed in *Lss^{f/f}* (n = 11) or *Lss^{wild/f}-K14* (n = 9) mice. No significant reduction in weight loss between *Lss^{f/f}* (n = 11) and *Lss^{wild/f}-K14* (n = 9) mice was observed. Each bar represents the mean ± SD. *P < 0.0001, P-values were calculated using Student's *t*-tests. **e,** Dye penetration assay on a newborn *Lss^{f/f}-K14* or *Lss^{f/f}* mouse. The experiments were repeated three times. **f,** Measurement of TEWL in newborn *Lss^{f/f}-K14* (n = 5) or *Lss^{f/f}* (n = 7) mice. P-values were calculated using Student's *t*-tests. **g,** TEM of the epidermis of newborn *Lss^{f/f}-K14* or *Lss^{f/f}* mice. Desmosomes were hypoplastic in the *Lss^{f/f}-K14* mice compared to those observed in *Lss^{f/f}* littermates. Scale bars = 100 nm. Data are from at least two independent experiments. P-values were calculated using Student's *t*-tests.

follicles and the thickness of the epidermis were not significantly different between the *Lss^/f*-*K14* and *Lss^{f/f}* mice. TEM revealed hypomorphic changes in the desmosomes of the *Lss^{f/f}*-*K14* mice (Fig 2g).

## Epidermal *Lss* ablation led to transient epilation in tamoxifen-inducible knockout mice

Next, we utilized a tamoxifen-inducible Cre/loxP system and generated *Lss flox/flox;Keratin14-Cre-ERT/+* (*Lss^{f/f}*-*K14ERT*) mice to avoid neonatal death. Epilation in the *Lss^{f/f}*-*K14ERT* mice occurred over 3 weeks after intraperitoneal injections of tamoxifen. In the following 1 to 2 weeks, the hair loss area peaked (Fig 3a and 3b). The fur gradually regrew, but it was not completely restored and remained sparse (Fig 3b). The surfaces of the ears or tails were reddish and rough (Fig 3b). Tamoxifen-treated *Lss^{f/f}* or untreated *Lss^{f/f}*-*K14ERT* mice did not exhibit any phenotype (Fig 3a). The ratios of (S)-2,3-epoxysqualene to lanosterol were significantly higher in the sebum of tamoxifen-treated *Lss^{f/f}*-*K14ERT* mice compared to the ratios in the tamoxifen-treated *Lss^{f/f}* or untreated *Lss^{f/f}*-*K14ERT* mice (Fig 3c). Histological analysis of the *Lss^{f/f}*-*K14ERT* mice revealed hypoplastic hair follicles, hyperplastic sebaceous glands, and thin root sheaths, which was consistent with the pathological results obtained from the patients (Fig 3d). The epidermises exhibited hyperplasia, parakeratosis, hyperkeratosis and liquefaction degeneration. Immunohistochemical analysis showed that the LSS protein levels decreased in the sebaceous gland and epidermal basal layer cells after tamoxifen treatment (Fig 3e). In contrast, the keratinocytes in the surface layer exhibited positive immunostaining for LSS. As the proportion of immunostained cells in the sebaceous glands decreased, the proliferation of the nearby epidermal cells increased (S11 Fig).

In addition, we topically administered 4-hydroxytamoxifen, which is an activated form of tamoxifen *in vivo*, on the skin of *Lss^{f/f}*-*K14ERT* mice or littermates. Consistent with the results of the intraperitoneal injections with tamoxifen, the 4-hydroxytamoxifen-treated areas exhibited transient epilation in *Lss^{f/f}*-*K14ERT* mice (S12 Fig). *Lss^{f/f}* (n = 2) and *Lss^{wild/f}*-*K14ERT* (n = 2) mice, which were the littermates of the treated *Lss^{f/f}*-*K14ERT* mice, did not show hair loss following topical administrations of 4-hydroxytamoxifen.

## Lens-specific *Lss* knockout mice had congenital cataracts

We created *Lss flox/flox;Pax6-Cre/+* (*Lss^{f/f}*-*Pax6*) mice in which the *Pax6* promoter was selected to create lens-specific knockout model mice [11,12]. The *Lss^{f/f}*-*Pax6* mice exhibited microphthalmia and small cloudy lenses that were not observed in the controls (Fig 4a–4d). Pathological analysis of the *Lss^{f/f}*-*Pax6* mice indicated lens swelling, degeneration, and liquefaction (Fig 4e–4h), which implied cataract changes. Immunohistochemical analysis demonstrated that the LSS protein levels were lower in the *Lss^{f/f}*-*Pax6* mice than they were in the *Lss^{f/f}* mice (Fig 4i–4l). The inner retina, which is the inner nuclear layer and ganglion cell layer, was thinner in *Lss^{f/f}*-*Pax6* mice than it was in *Lss^{f/f}* mice (S13 Fig). In particular, there were fewer retinal ganglion cells in *Lss^{f/f}*-*Pax6* mice than there were in *Lss^{f/f}* mice. (S)-2,3-epoxysqualene, lanosterol, squalene, and cholesterol were not detected in the lenses of *Lss^{f/f}* or *Lss^{f/f}*-*Pax6* mice.

## Discussion

Here, we describe two sibling patients with biallelic *LSS* loss-of-function mutations that led to compromised LSS protein expression and an increased ratio of substrate to product in the patients' sebum, indicating LSS enzymatic inhibition *in vivo*. The LSS enzyme disruption was also reproducible in the skin sebum of inducible epidermis-specific *Lss* knockout mice. Previous studies could not demonstrate LSS dysfunction by analysis of blood samples [7,8].

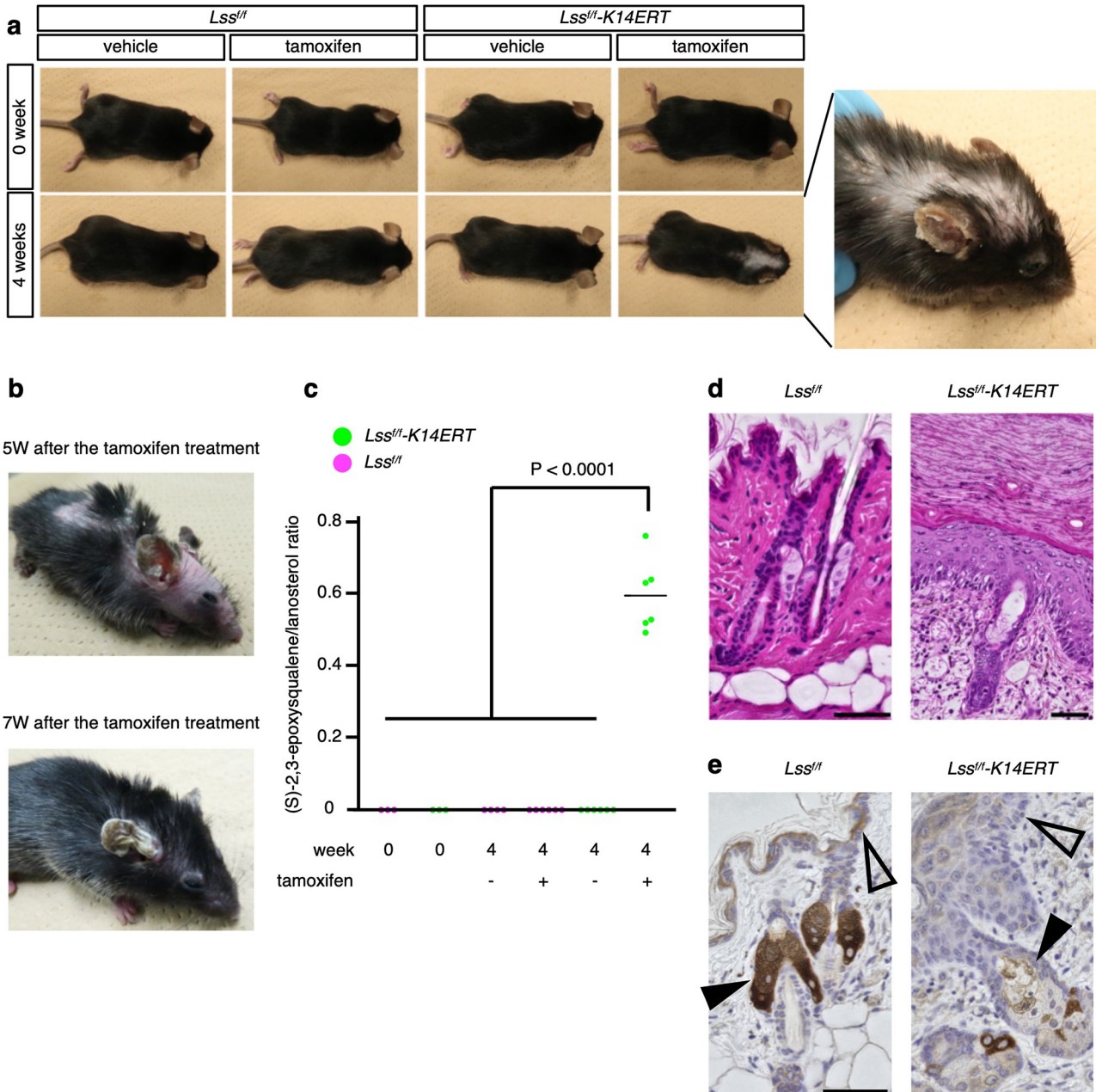

**Fig 3. Inducible epidermis-specific deletion of *Lss* caused alopecia that was spontaneously and partially reversed in the mice. a**, Gross appearance of the *Lss^{f/f}-K14ERT* or *Lss^{f/f}* mice pre- and postadministration of intraperitoneal tamoxifen. *Lss^{f/f}-K14ERT* mice 4 weeks after tamoxifen treatment, which affected baldness from the head to the back. The experiments were repeated at least three times. **b**, The peak of hair loss in *Lss^{f/f}-K14ERT* mice was 5 weeks after the tamoxifen treatment (top). The hair had spontaneously regrown 7 weeks after the treatment (bottom). W, weeks. **c**, Skin sebum analysis of *Lss^{f/f}-K14ERT* (n = 15) or *Lss^{f/f}* (n = 13) mice pre- and postadministration of intraperitoneal tamoxifen or vehicle. At 6 weeks of age, (S)-2,3-epoxysqualene was not detected in either *Lss^{f/f}-K14ERT* (n = 3) or *Lss^{f/f}* (n = 3) mice. After 4 weeks of treatment, the ratios of (S)-2,3-epoxysqualene to lanosterol in the sebum of tamoxifen-treated *Lss^{f/f}-K14ERT* mice (n = 6) were significantly elevated compared to those of vehicle-treated *Lss^{f/f}* (n = 4), tamoxifen-treated *Lss^{f/f}* (n = 6), or vehicle-treated *Lss^{f/f}-K14ERT* (n = 6) mice. P-values were calculated using Tukey-Kramer tests. NS, not significant. **d**, H&E staining and **e**, LSS immunohistochemistry of skin from *Lss^{f/f}-K14ERT* or *Lss^{f/f}* mice 4 weeks after tamoxifen treatment. The epidermis of *Lss^{f/f}-K14ERT* mice showed severe acanthosis compared to *Lss^{f/f}* mice. LSS protein expression was decreased in the epidermal basement membrane (open arrowhead) and sebaceous cells (closed arrowhead) of the *Lss^{f/f}-K14* mice compared to the expression in *Lss^{f/f}* mice. Scale bars = 100 nm.

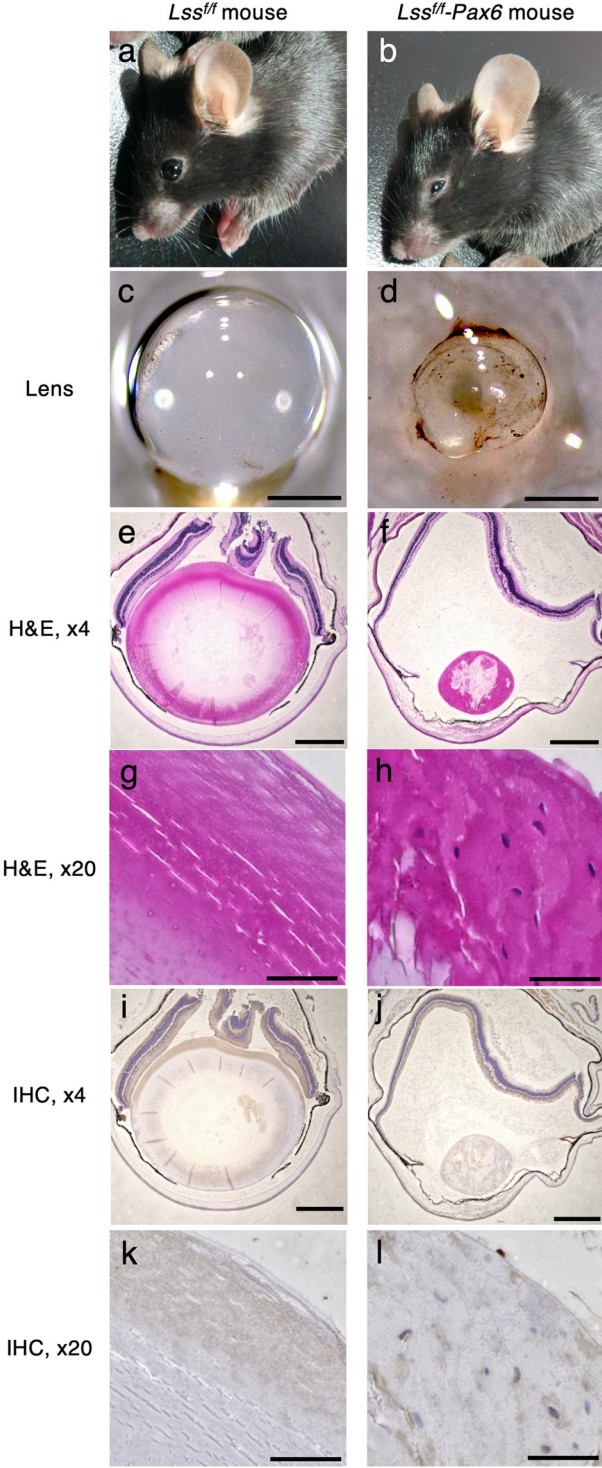

**Fig 4. Lens-specific deletion of *Lss* caused cataracts in mice. a-d**, Images of *Lss^{f/f}*-*PAX6* or *Lss^{f/f}* mice (14 weeks of age) and their lenses. **e-l**, H&E staining and immunohistochemistry of the lenses of *Lss^{f/f}*-*PAX6* or *Lss^{f/f}* mice. These experiments were repeated at least three times. Scale bars = 500 μm (b), 50 μm (c).

Measurement of cholesterol intermediates from skin, an affected organ, but not blood, allowed us to validate the LSS enzymatic block. The ratio of (S)-2,3-epoxysqualene to lanosterol in the sebum would be a good biomarker for the diagnosis of LSS deficiency. Hypotrichosis and cataracts were reproduced by creating tissue-specific *Lss* knockout mice. Our results confirmed that hypotrichosis and cataracts, the core symptoms in patients with biallelic *LSS* mutations, are caused by LSS enzymatic deficiency in each tissue, not by circulating abnormal metabolites in the blood.

We recapitulated hypotrichosis and cataracts by creating tissue-specific *Lss* knockout mice, and these outcomes are caused by knocking out *Lss* in the respective organs. Both of these symptoms are also associated with other inherited disorders of cholesterol biosynthesis. Hypotrichosis is observed in other conditions associated with deficiencies of cholesterol synthesis enzymes, including Greenberg dysplasia, SC4MOL deficiency, X-linked dominant chondrodysplasia punctata-2 (CDPX2), and congenital hemidysplasia with ichthyosiform erythroderma and limb defects (CHILD) syndrome [13–17]. Cataracts are also known to develop in patients with deficiencies of cholesterol synthesis enzymes, such as SC4MOL deficiency, CDPX2, lathosterolosis and Smith-Lemli-Opitz syndrome (SLOS) [16,18–20]. However, given that most mice that have had cholesterol synthesis genes knocked out are embryonic lethal [21], conventional knockout mouse studies are not feasible for recapitulation of the symptoms in patients with these deficiencies. Moreover, it is not understood whether symptoms of patients with abnormal cholesterol metabolism are local events or are secondary effects caused by enzyme deficits. Our study validated that local abnormalities in cholesterol metabolism can cause symptoms in each tissue using conditional knockout mouse models; the results are similar to those from a previous mouse study of Insig-1/2, which is involved in a feedback system of cholesterol synthesis [22].

Tamoxifen-induced *Lss* knockout in the mouse epidermis led to epilation. The hyperkeratosis observed in the epidermis was consistent with the ichthyosis observed in the patient during the neonatal period (S2 Fig) and with the histological analysis of the patient's skin (Fig 1c and 1d). Recent studies also demonstrated similar dermatological findings, including ichthyosis, in other patients with biallelic *LSS* mutations [8]. As with hypotrichosis, hyperkeratosis also occurs in other disorders of cholesterol biosynthesis [13–17]. Interestingly, the epilation of the tamoxifen-treated mice was transient. A possible explanation for the hair regrowth is that the surrounding cells might supply LSS-deficient cells with deficient products or consume excess substrates. Consistent with this hypothesis, a severe phenotype was observed in parts of the body with few subcutaneous tissues, such as the ears (Fig 3b) and tail. Contrary to the tamoxifen-inducible system, the phenotypes of LSS deficiency were not fully reproduced in the *Lss^f/f^-K14* mice due to neonatal lethality. The mice failed to develop normal skin, which especially affected the skin barrier function. The failure to develop a skin barrier is consistent with the mouse knockout of *DHCR24*, which catalyzes cholesterol intermediates [23]. Of note, in our experiments, only one mouse grew to the adult stage (S14 Fig). The mouse grew normally and could not be differentiated from the other littermates until weaning but gradually exhibited baldness and affected skin scales after approximately 6 weeks of age. Unlike the tamoxifen-treated *Lss^f/f^-K14ERT* mice, alopecia was not recovered in this mouse until the mouse died of unknown causes at 10 weeks of age. Considering that the hair follicles of epidermis-specific knockout mice developed normally (S11 Fig), cell-autonomous cholesterol biosynthesis by LSS enzyme might be crucial not for morphogenesis and primary hair development but for secondary hair germ in hair development. By preventing water evaporation from the skin, *Lss^f/f^-K14* mice might avoid neonatal lethality. If the *Lss^f/f^-K14* mice could survive and reach adulthood, these mice would be excellent animal models for clarifying the pathophysiology related to disrupted *Lss* and investigating treatments.

The pathogenesis of LSS deficiency is unknown, but the mechanism is likely attributed to aberrant SHH or WNT signaling. Although the association of *LSS* mutations and holoprosencephaly was not proven by pedigree analysis in a previous study [24], some patients with *LSS* mutations, including our patients, have midline anomalies or growth failure [8], which are observed in diseases of cholesterol metabolism or SHH signaling [25,26]. SLOS, which impairs cholesterol synthesis due to DHCR7 deficiency and often produces similar symptoms, has been proposed to be involved in the SHH pathway [20,27]. WNT signaling, which interacts with SHH-related molecules during embryogenesis in several tissues, is also required for hair follicle generation [28]. Thus, we surveyed whether LSS deficiency caused Shh or Wnt dysregulation. However, there were no significant differences in Shh or Wnt transcriptional responses against the exogenous Shh or Wnt protein between *Lss*-depleted and nondepleted MEFs (S15 Fig). These results might imply that the underlying mechanism is not simply caused by the upregulation or downregulation of these signaling pathways because spatial and gradient SHH or WNT signaling seems to be key in skin formation [29]. Moreover, the reason why each patient with biallelic *LSS* mutations shows different combinations of phenotypes, such as hypotrichosis and/or cataracts, is also unknown [5–8]. Further experimental studies are required to understand the pathogenesis and mechanism of the phenotypic variation in LSS deficiency.

In conclusion, our study verified that biallelic *LSS* mutations caused LSS enzymatic dysfunction in skin through analysis of sebum. Our mouse studies underscored that hypotrichosis and cataracts are caused by LSS enzymatic dysfunctions in each tissue. Local treatment in LSS deficiency might alleviate symptoms; patients with CHILD syndrome, a deficiency of downstream intermediates of LSS, was successfully treated by topical application of statin or cholesterol plus statin [15,30]. The mouse model of LSS deficiency might serve as a pathophysiological research model and aid in the development of novel treatments.

## Materials and methods

### Sequencing analysis

Whole-exome sequencing was performed as described elsewhere [31] but with minor modifications. Briefly, genomic DNA was purified using a SureSelectXT Human All Exon V5 (50 Mb) Kit (Agilent Technologies, Santa Clara, CA, USA) and was sequenced on a HiSeq2000 (Illumina, San Diego, CA, USA) with 101-base-pair paired-end reads. The reads were mapped to the human genome build UCSC hg19 using Novoalign v. 2.08.05 (http://www.novocraft. com), and single-nucleotide variants (SNVs) and insertions and/or deletions (indels) were identified using the Genome Analysis Toolkit (GATK) v. 1.6–13. After quality filtering, variants were annotated using ANNOVAR. Nonsynonymous SNVs, splice-site variants and indels were retained. We excluded variants present at <1% minor allele frequency (MAF) in the 1000 Genomes database, the ExAC database version 0.3 (all and East Asian populations) and the HGVD. Sanger sequencing was performed to validate the variants identified by whole-exome sequencing.

### Plasmids

Wild-type, R234fs, and R177Q human *LSS* cDNAs (NM_002340.5, carrying SNP rs2254422 and rs2254524) were prepared by RT-PCR with RNA from the blood of the patients and their family members. cDNAs were subcloned into pCMV6-AN-DDK (PS100014, OriGene, Rockville, MD, USA). The final constructs were verified by sequencing.

## Cell culture and transfection

HeLa cells (RCB0007, provided by the RIKEN BRC through the National BioResource Project of the MEXT, Japan) were cultured in Dulbecco's modified Eagle's medium (DMEM) containing low levels of glucose, which was supplemented with 10% fetal bovine serum and 1% penicillin and streptomycin (100 U/ml and 100 μg/ml, respectively); cells were maintained at 37˚C in a 5% $CO_2$ incubator. The pCMV6-FLAG-LSS (wild-type, R234fs, and R177Q) plasmid was transfected into HeLa cells using Lipofectamine LTX (15338100, Thermo Fisher Scientific Inc., MA, USA). Twenty-four hours after transfection, cell lysates were harvested and subjected to immunoblot analysis.

## Western blot analysis

Proteins extracted from PBMCs or HeLa cells expressing FLAG-LSS were subjected to immunoblot analysis. Mouse anti-FLAG (F3165, Sigma-Aldrich, Merck Millipore), rabbit anti-LSS (13715-1-AP, Proteintech, Rosemont, IL, USA), HRP-conjugated anti-mouse IgG (NA9310V, GE Healthcare UK Ltd., Buckinghamshire, England), HRP-conjugated anti-rabbit IgG (NA9340V, GE Healthcare UK Ltd.), and peroxidase-conjugated anti-β-actin (017–24573, Wako, Osaka, Japan) antibodies were used.

## Sample collection and preparation for LC-MS/MS analysis

Human sebum was collected using acetone-soaked cotton pads as described previously [32], and it was extracted with acetone. Mouse sebum was collected similarly and was syringe filtered (GL Chromatodisc 25N, GL Sciences, Tokyo, Japan). Mouse skin was sampled as described previously [33]. Mouse skin homogenates were prepared using 0.9% KCl containing 1 mM ethylenediaminetetraacetic acid. Total lipids from the above samples and human serum were extracted by the Folch method as described previously [34,35] but with modifications. All samples were dissolved in methyl *tert*-butyl ether (MTBE)/methanol (2:1) prior to analysis.

## LC-MS/MS analysis

Samples were analyzed with a QTRAP 4000 mass spectrometer equipped with an Exion HPLC system (SCIEX, Tokyo, Japan). Samples were eluted from a C30 column (YMC Carotenoid 250×4.6 mm, YMC, Kyoto, Japan) at a flow rate of 1.0 mL/min with a binary gradient consisting of solvent A (methanol/MTBE/water, 83:15:2) and solvent B (methanol/MTBE/water, 8:90:2). The gradient profile was as follows: 0–10 min, 0–50% B, linear; 10–20 min, 100% B; 20–25 min, 0% B. The column temperature was maintained at 40˚C. Compounds were detected with atmospheric-pressure chemical ionization in positive ion mode, and they were identified through comparison of peaks between the samples and commercial standards. Compounds were quantified with external calibration curves using the corresponding standards. Further MS conditions are described in S5 Table.

## Mice

Cryopreserved sperm of *Lss* tm1a knockout mice were purchased from the European Mouse Mutant Archive. Heterozygous *Lss* tm1a mice were generated at the Institute for Animal Experimentation, Tohoku University Graduate School of Medicine, and then they were crossed with *Flpe* mice (RBRC01834, RIKEN BRC, Japan) [36] to obtain *Lss wild/flox* mice that had a conditional allele (S8 Fig). *Lss wild/flox* mice were bred with *K14-Cre* transgenic mice (stock no. 004782; The Jackson Laboratory, USA) to generate epidermis-specific knockout mice. *Lss^{f/f}-K14ERT* or *Lss^{f/f}-Pax6* mice were similarly created using *K14-CreERT*

transgenic mice (stock no. 005107; The Jackson Laboratory, USA) or *PAX6-Cre* transgenic mice (stock no. 024688; The Jackson Laboratory, USA), respectively. All mice in these studies were on a C57BL/6 background except the *K14-CreERT* mice, which were a strain of ICR mice backcrossed with C57BL/6 mice for at least 5 generations. Day 0.5 of pregnancy (E0.5) was defined as the morning on the day after overnight mating. Mouse tail genomic DNA was extracted with a Maxwell 16 Mouse Tail DNA Purification Kit (Promega, USA). Mouse genotypes were determined by probe-based PCR analysis. The PCR primers (Nihon Gene Research Laboratories, Miyagi, Japan) and probes (Integrated DNA Technologies, Coralville, IA, USA) are described in S6 Table. Tamoxifen (T5648, Sigma-Aldrich, USA) was dissolved in corn oil, and 1 mg was injected intraperitoneally daily for 5 consecutive days. 4-Hydroxytamoxifen (H954725, Toronto Research Chemicals, Toronto, Canada) was dissolved in 100% acetone at 4 mg/dl, and 0.4 mg was topically applied daily for 5 consecutive days.

### qPCR analysis

The mouse epidermises were isolated after allowing the skin to float on trypsin for 24 hours. Epidermis and liver were homogenized using a pellet pestle cordless motor (Kimble, Vineland, NJ, USA). RNA was extracted with RNeasy (QIAGEN, Hilden, Germany), and reverse transcription was performed with a PrimeScript RT Reagent Kit with gDNA Eraser (TakaraBio, Shiga, Japan). cDNA was amplified with TaKaRa Ex Taq (RR001A, TaKaRa Bio Inc., Otsu, Japan) and PrimeTime qPCR probe assays (Integrated DNA Technologies, Coralville, IA, USA). qPCR primers and probes for *Lss* (Mm.PT.58.41802559) and *Gapdh* (Mm.PT.39a.1) were used. qPCR reactions were performed using StepOnePlus (Sigma-Aldrich, St. Louis, MO, USA) with standard conditions.

### Histology and immunohistochemistry

Skin samples were fixed in formalin, embedded, and cut into 5-μm sections. HE staining was performed using standard procedures. For immunostaining, slides were incubated with an anti-LSS antibody according to the manufacturer's instructions (13715-1-AP, Proteintech, Rosemont, IL, USA). Images were captured with a BIOREVO BZ-9000 microscope (Keyence, Osaka, Japan).

### Electron microscopy

Mouse skin samples were fixed with 2% paraformaldehyde and 2.5% glutaraldehyde in 0.1 M cacodylate buffer overnight at 4˚C. After being cut into small pieces, the specimen was post-fixed in 1% osmium tetroxide for 90 min on ice, rinsed in distilled water, successively dehydrated in 50, 60, 70, 80, 90, and 95% ethanol for 10 min each (except for 70% ethanol, which was used for 25 min). Then, specimens were incubated in for 20 minutes 100% ethanol (3 times), treated with propylene oxide for 10 min (2 times), and finally, embedded in epoxy resin. Ultrathin (80 nm) sections were prepared on an ultramicrotome (EM UC-7, Leica, Heerbrugg, Switzerland) with a diamond knife, and then slides were stained with 2% uranyl acetate and 1% lead citrate and were viewed using an electron microscope (H-7600, Hitachi, Tokyo, Japan).

### Skin barrier function assays

A dye penetration assay was performed as described previously [37]. Briefly, newborn mice were rinsed in PBS and successively dehydrated in 25, 50, 75, and 100% methanol for 1 min each. Mice were then rehydrated with PBS, stained with 0.05% toluidine blue in PBS for 10

min, destained with PBS for 20 min, and photographed. All steps were performed at room temperature. TEWL was recorded using AquaFlux (Biox, London, UK). Measurements were performed after calibrating the device at room temperature.

## Statistical analysis

Statistical analysis was performed with JMP Pro 13.0. P-values <0.05 were considered statistically significant.

## Ethics statement

Approval for human subject research was obtained from the ethics committee of Tohoku University Graduate School of Medicine (approval number: 2015-1-694). Written informed consent was obtained from all participants in the study. Approval for animal experiments was provided by the Animal Care Facility of the Tohoku University Graduate School of Medicine (approval number: 2016MdA-371, 2016MdA-372).

## Supporting information

**S1 Fig. The metabolic pathway of cholesterol biosynthesis.** Cholesterol is synthesized from acetyl CoA via the Bloch or Kandutsch-Russell pathway. In this synthesis pathway, LSS converts (S)-2,3-epoxysqualene to lanosterol. The abbreviations are as follows: FDFT1, farnesyldiphosphate farnesyltransferase 1; ff-MAS, follicular fluid meiosis-activating sterol; t-MAS, testis-meiosis activating sterol; LBR, lamin B receptor; SC4MOL, sterol C4-methyloxidase-like; NSDHL, NAD(P)H steroid dehydrogenase-like protein; HSD17B7, 17-beta-hydroxysteroid dehydrogenase; EBP, emopamil-binding protein; SC5DL, sterol C5-desaturase; DHCR7, 7-dehydrocholesterol reductase; and DHCR24, 24-dehydrocholesterol reductase.
(TIF)

**S2 Fig. Patient II-2 at 1 month of age.** The patient exhibited ichthyosis-like scales on the skin of his chest, abdomen and upper extremities but not on his face and head.
(TIF)

**S3 Fig. Sagittal T1-weighted head MRI images of patients II-2 and II-3.** Sagittal T1-weighted head MRI shows partial agenesis of the corpus callosum (from the posterior midbody to splenium) in patient II-2, while the MRI shows a normal corpus callosum in patient II-3.
(TIF)

**S4 Fig. Transmission electron microscopy images of the skin of patient II-3.** (a) Hair matrix of the patient. The number of melanin granules was small. The inner root sheath was thin. (b) The basement layer cells were tightly adhered to each other. (c) The desmosome was well developed. Scale bars = 200 μm (a), 50 μm (b), and 2.5 μm (c).
(TIF)

**S5 Fig. Conservation of a missense variant in LSS.** The R177Q residue is conserved in organisms from humans to zebrafish. LSS homologs were aligned with Clustal Omega.
(TIF)

**S6 Fig. The frameshift variant c.701_716del in *LSS* leads to NMD.** EBV-LCLs from patient II-2 and II-3 were treated with 100 μg/mL cycloheximide (037–20991, FUJIFILM Wako Pure Chemical Corporation, Osaka, Japan) for 4 hours to suppress NMD. Total RNA was extracted with an RNeasy Kit (Qiagen Inc., Valencia, CA, USA). RT-PCR was performed using a PrimeScript™ II High Fidelity RT-PCR Kit (TaKaRa, Shiga, Japan) to amplify the cDNA surrounding

the c.701_716del mutation, which was followed by Sanger sequencing.
(TIF)

**S7 Fig. Mass spectra was collected by LC-MS/MS for the plasma from the patients or family members.** Blue arrows indicate peaks of 2,3-oxidosqualene. Lanosterol could not be detected by this analysis.
(TIF)

**S8 Fig. Diagrams of the knockout allele.** The tm1a allele was a knockout first allele, and it was converted to a conditional allele, tm1c (flox), by FLP-mediated recombination. The tm1c (flox) allele was converted to a deletion allele by Cre-mediated recombination.
(TIF)

**S9 Fig. Cholesterol quantification in newborn *Lss^f/f^-K14* or *Lss^f/f^* skin.** Mouse skin was gravimetrically weighed, and homogenates were prepared using 0.9% KCl containing 1 mM ethylenediaminetetraacetic acid. Total lipids were extracted from the homogenates using the Folch method [34, 35], and the extract was dissolved in 100 μL of 2-propanol. Human sebum was collected using acetone-soaked cotton pads [32], and it was dissolved in 200 μL of 2-propanol. The total cholesterol content in each sample was measured enzymatically with a commercial kit (Cholesterol E; Wako Pure Chemical Industries, Ltd., Osaka, Japan) according to the manufacturer's protocol. Each bar represents the mean ± SEM (n = 5 per group). P-values were calculated using Student's *t*-tests. NS, not significant.
(TIF)

**S10 Fig. Light microscopic images of *Lss^f/f^-K14* and *Lss^f/f^* mouse skin.** (a) Histological analysis did not show macroscopic differences in epidermal structure between the *Lss^f/f^-K14* and *Lss^f/f^* mice. (b) Immunohistochemistry showed that expression of the LSS protein was decreased in the epidermal basement membrane (open arrowhead) and hair follicles (closed arrowhead) of the *Lss^f/f^-K14* mice compared to the *Lss^f/f^* littermates. Scale bars = 100 μm.
(TIF)

**S11 Fig. Immunohistochemical analysis of skin from *Lss^f/f^-K14ERT* mice with alopecia.** Keratosis is mild in epidermal areas near immunostained sebaceous cells (a, b). In contrast, severe keratosis is found in epidermal areas lacking immunostained sebaceous cells (c). Scale bars = 100 μm.
(TIF)

**S12 Fig. Topical tamoxifen-treated *Lss^f/f^-K14ERT* mice.** Images of *Lss^f/f^-K14ERT* mice arranged chronologically after topical application of 4-hydroxytamoxifen, which was applied within the black rectangle. Depilation gradually occurred after 3 weeks of 4-hydroxytamoxifen treatment. The depilation area was greatest after 4 weeks. Similar to the effects of intraperitoneal injection of tamoxifen, the hair gradually regrew. Gray hairs were mixed among the hairs in the recovered area.
(TIF)

**S13 Fig. Microscopic analysis of the retina in *Lss^f/f^-Pax6* and *Lss^f/f^* mice.** The inner retina, which is the inner nuclear layer and ganglion cell layer, was thin in *Lss^f/f^-Pax6* mice compared to *Lss^f/f^* mice. There were fewer retinal ganglion cells in *Lss^f/f^-Pax6* mice than there were in *Lss^f/f^* mice. In contrast, there was no obvious difference in the thickness of the outer nuclear layers between *Lss^f/f^-Pax6* and *Lss^f/f^* mice. These pictures were captured using an Olympus BX53 microscope (Olympus, Tokyo, Japan). A 10x objective lens was used. Scale bars = 50 μm.
(TIF)

**S14 Fig. The surviving *Lss^{f/f}-K14* mouse.** Sequential images of an adult *Lss^{f/f}-K14* mouse (a–d). At weaning, the mouse could not be differentiated from the other littermates. The mouse gradually lost hair after 6 weeks of age (a). Desquamation started at 9 weeks of age (c). A partial skin defect on the head with oozing was observed. The mouse died due to unknown causes at 10 weeks of age (d). Analysis of histological specimens from the dead mouse revealed severe hyperkeratosis, dyskeratosis, and hypertrophic sebaceous cells (e). Scale bars = 100 μm. (TIF)

**S15 Fig. Transcriptional responses against the exogenous Shh or Wnt protein between *Lss*-depleted and nondepleted MEFs.** We measured the transcriptional responses of MEFs against the exogenous Shh or Wnt protein as previously described [27, 38] with minor modifications. Embryos (E13.5–14.5) were harvested by aseptic techniques from pregnant *Lss^{f/f}* mice that had been mated with *Lss^{f/f}* male mice. The heads, extremities, and tails were cut off, and the livers were removed. The remaining body parts were minced by scissors and transferred to a 50 mL tube with PBS. The tube was centrifuged at 200 g for 5 min. The supernatant was aspirated, and the pellet was resuspended with 1 mL 0.25% trypsin-EDTA per embryo. The suspension was incubated for 15 min in a 37˚C water bath with shaking. Fetal bovine serum was added (the same volume as trypsin) to the suspension. After letting the suspension sit for several minutes and allowing the embryo fragments sank to the bottom of the tube, the serous part was removed first and filtered, and then the remaining thick part was passed through a 100 μm filter. The filtered solution was centrifuged at 200 g for 5 min. The supernatant was aspirated, the pellet was resuspended in DMEM, and the suspension was transferred to a dish. MEFs were plated at $5 \times 10^4$ cells in 250 μL of DMEM per 48-well plate and transduced with AAV1 or a mock control (105548-AAV1 or 105537-AAV1, respectively, Addgene, Cambridge, MA, USA) at a multiplicity of infection of $5 \times 10^5$. The next day, 250 μL of DMEM was added to each well. The following day, 2 μg/mL Shh (464-SH-025/CF, R&D systems, Minneapolis, MN, USA) or 100 ng/mL Wnt-3a (1324-WN, R&D systems, Minneapolis, MN, USA) were added to the medium. Three days after transduction, total RNA was extracted with RNeasy (QIAGEN, Hilden, Germany), and reverse transcription was performed with a PrimeScript RT Reagent Kit with gDNA Eraser (TakaraBio, Shiga, Japan). cDNA was amplified with TaKaRa Ex Taq (RR001A, TaKaRa Bio Inc., Otsu, Japan) and PrimeTime qPCR probe assays (Integrated DNA Technologies, Coralville, IA, USA). qPCR primers and probes for *Gli1* (Mm.PT.58.11933824), *Axin2* (Mm.PT.58.8726473), *Lss* (Mm.PT.58.41802559), and *Gapdh* (Mm.PT.39a.1) were used. qPCR reactions were cycled using StepOnePlus (Sigma-Aldrich, St. Louis, MO, USA) with standard conditions. (TIF)

**S1 Table. Variant filtering strategy.**
(XLSX)

**S2 Table. Mutations in *LSS*.**
(XLSX)

**S3 Table. Laboratory results of the patients with LSS biallelic mutations and unaffected family members in this study.**
(XLSX)

**S4 Table. Genotype distribution of neonatal mice born from *Lss^{wild/flox}-Keratin14* male and *Lss^{flox/flox}* female mice.**
(XLSX)

**S5 Table. MS parameters used for LC-MS/MS analysis.**
(DOCX)

**S6 Table. Primers and probes.** The forward primers of *Lss tm1a*, *Lss tm1c*, and *Lss tm1d* were the same. The existence of the *Pax6-Cre* allele was confirmed using generic Cre primers and a probe.
(DOCX)

## Acknowledgments

The authors would like to thank the patients, families and healthy volunteers who participated in this study. We thank Yoko Chiba, Kumi Ito, Miyuki Tsuda, Mami Kikuchi, Makiko Nakagawa, Yoko Tateda, and Kiyotaka Kuroda for providing technical assistance.

We also acknowledge the technical assistance of the Biomedical Research Core of the Tohoku University Graduate School of Medicine and the Biomedical Research Unit of Tohoku University Hospital. We deeply appreciate Keisuke Nishio and Prof. Ichiro Miyoshi for recovering cryopreserved mouse sperm. We thank Naomi Sato for expert pathology-related advice.

## Author Contributions

**Conceptualization:** Yoichi Wada, Atsuo Kikuchi, Junya Ito, Shigeo Kure.

**Funding acquisition:** Atsuo Kikuchi, Shigeo Kure.

**Investigation:** Yoichi Wada, Atsuo Kikuchi, Naoki Shimizu, Junya Ito, Ryo Onuma, Fumiyoshi Fujishima, Eriko Totsune, Ryo Sato, Tetsuya Niihori, Kota Sato, Toru Nakazawa, Setsuya Aiba, Kiyotaka Nakagawa.

**Methodology:** Naoki Shimizu, Junya Ito, Ryo Onuma, Tetsuya Niihori, Matsuyuki Shirota, Ryo Funayama, Keiko Nakayama, Yoko Aoki, Kiyotaka Nakagawa.

**Project administration:** Shigeo Kure.

**Resources:** Atsuo Kikuchi, Akimune Kaga, Fumiyoshi Fujishima, Setsuya Aiba.

**Supervision:** Atsuo Kikuchi, Kiyotaka Nakagawa, Shigeo Kure.

**Validation:** Yoichi Wada.

**Visualization:** Yoichi Wada, Atsuo Kikuchi, Kota Sato.

**Writing – original draft:** Yoichi Wada, Atsuo Kikuchi, Naoki Shimizu, Junya Ito.

**Writing – review & editing:** Yoichi Wada, Atsuo Kikuchi, Toru Nakazawa, Shigeo Kure.

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
