## [Decision Letter · Decision Letter 0]

23 Sep 2019

Dear Dr Kikuchi,

Thank you very much for submitting your Research Article entitled 'Metabolic and pathologic profiles of human LSS deficiency recapitulated in mice' to PLOS Genetics. Your manuscript was fully evaluated at the editorial level and by independent peer reviewers. The reviewers appreciated the attention to an important problem, but raised some substantial concerns about the current manuscript. Based on the reviews, we will not be able to accept this version of the manuscript, but we would be willing to review again a much-revised version. We cannot, of course, promise publication at that time.

If you decide to revise the manuscript for further consideration at PLOS Genetics, please aim to resubmit within the next 60 days, unless it will take extra time to address the concerns of the reviewers, in which case we would appreciate an expected resubmission date by email to plosgenetics@plos.org.

[LINK]

We are sorry that we cannot be more positive about your manuscript at this stage. Please do not hesitate to contact us if you have any concerns or questions.

Yours sincerely,

Luke Engelking

Guest Editor

PLOS Genetics

Gregory Barsh

Editor-in-Chief

PLOS Genetics

All three reviewers were enthusiastic about the major findings of this manuscript. However, a number of issues were identified that will require revision and/or clarification.  These issues will need to be addressed in a point-by-point rebuttal.  Lastly, the following issues in particular may require additional experimental data:

- Photomicrographs of the hair and ocular structures in addition to the lens

- Measurements of sterol intermediates

- Gene and or protein expression data exploring which signaling pathways, such as Wnt or Hh-related pathways, are impacted by LSS deficiency

- Inclusion of the suggested controls for the transfection experiments in hela cells

- Methods used for lanosterol and 2,3-epoxysqualene quantification

Reviewer's Responses to Questions

**Comments to the Authors:**

Reviewer #1: The authors submit an interesting study whereby they describe two new patients with novel biallelic mutations in lanosterol synthetase (LSS) who exhibit hair and midline structural abnormalities and a concomitant increase in the substrate:product (S-2,3-epoxysqualene:lanosterol) ratio secondary to an enzymatic block. Additionally, the authors generate tissue-specific LSS-deficient mouse lines with hair/skin and lens abnormalities; the skin from these tissue-specific LSS-deficient mice demonstrate increased S-2,3-epoxysqualene:lanosterol ratios which may alter normal skin homeostasis and hair follicle cycling. The authors provide extensive data throughout the manuscript; however, there are some concerns:

1) In Figure 1g, the authors illustrate the S-2,3-epoxysqualene:lanosterol ratio from the sebum of the affected patients, their unaffected family members, and controls. Given that cholesterol is essential in the formation of an intact skin barrier, it would be beneficial to show the values of all sterol intermediates and cholesterol in the sebum in these groups since one can speculate that these skin and hair abnormalities may be due to either a deficiency of cholesterol, a buildup of toxic intermediates, or both.

2) In addition to the high magnification photomicrograph in Figure 1e, a lower magnification photomicrograph of the hair follicles is needed to assess the extent of hair follicle pathology.

3) As stated above in Concern 1, values for the remainder of the sterol intermediates and cholesterol would be useful in explaining the pathophysiology seen in the LSSf/f-K14 mice illustrated in Figure 2.

4) Have the authors considered treating the 4-week-old tamoxifen-treated LSSf/f-K14ERT mice that show hair loss and skin abnormalities with statins and/or cholesterol + statins? As the authors suggest in the Discussion, this therapy may reduce these intermediates and reverse the pathology seen in these mice, particularly since the hair follicle cycles continuously throughout the majority of a mammal’s lifetime.

5) Figure 4 depicts the lens pathology seen in mice with LSS deficiency restricted to the lens. From the photomicrographs shown in Figure 4b, there appears to be pathology within other ocular/intraocular structures, such as within the cornea and retina, than in just the lens. Given Pax6-Cre is expressed during early eye development, loss of LSS within other ocular/intraocular tissues may not be surprising. It would be interesting to determine if the LSSf/f-Pax6 mice have retinal or other ocular/intraocular abnormalities besides those of the lens. Lastly, the authors do not state the age of these mice and do not show the LC-MS/MS data of lens or other ocular structures.

6) Additional points include:

-Line 135: Is the variant supposed to be p.Arg177Gln instead of p.Arg177Glu?

-Line 226: Should it be “increased ratio of substrate to product” instead of decreased?

-The flow of the overall manuscript is rather choppy at times and could use proofreading by a native English speaker for better flow and readability.

Overall, the manuscript is interesting and not only provides more clarification of the tissue-specific pathophysiology seen in these malformation syndromes but also hints at diagnostic assays/therapeutic interventions for LSS deficiency.

Reviewer #2: In the manuscript entitled "Metabolic and pathologic profiles of human LSS deficiency recapitulated in mice ", the authors demonstrate disruption of lanosterol synthetase in an inducible mouse model of LSS deficiency results in biochemical deficits and tissue specific changes present within affected human subjects. The work presented within the manuscript is important to the fields of cholesterol biosynthesis defects, rare disease research, and epidermal function. While the authors nicely demonstrate the identification of defined mutations within a small patient cohort of LSS deficiency and recapitulate some of the tissue deficits within a newly derived mouse model, the data presented are not likely of broad enough scientific significance for acceptance in PLOS Genetics. A significant advancement forward for LSS deficiency, such as identification of causative pathways downstream of LSS mutation or amelioration of pathological findings, would greatly enhance the significance of the manuscript.

Concerns related to the manuscript are detailed below.

1. While the methodology utilized within the manuscript is solid, the data presented, including generation of an inducible Lss mouse model, is largely descriptive. While the inducible Lss mouse is an exciting development for elucidating the pathological mechanisms subsequent to Lss disruption, the addition of mechanistic data is really needed. Are there specific signaling pathways subsequent to Lss deletion that are pathogenic?

2. It’s not clear the significance of the experiments detailing FLAG-tagged LSS mutations and changes in protein expression of specific mutants, particularly in HeLa cells that are demonstrated to already express LSS protein. How a transient transfection of a plasmid containing a mutant cDNA for LSS leads to decreased expression of native LSS is confusing. Further, why specific mutations affect this versus others is similarly confusing. Controls for this experiment for expression levels of the plasmids in an LSS null cell line would be critical.

3. The clinical findings from the two subjects identified are similar but not identical to the K14 mouse model used. The presence of cleft palate, midline anomalies, and corpus collosum agenesis in particular would not be affected by K14-Cre mediated deletion of Lss. Are these deficits present in the newly created Lss mutant mouse if a constitutive or embryonic Cre-driver used?

4. The quantitation and detection of sterols is not clearly defined. I believe there are typos within the methods, including what temperature was utilized for compound ionization. Further, how the compounds of interest were identified (i.e. through comparison to a known standard?) was performed. The number of samples analyzed as biological vs. technical replicates is also not clear. Further, the explanation regarding detection of lanosterol and 2,3-epoxysqualene in plasma vs. sebum of patients is unclear. It’s not clear why lanosterol would be detected.

5. For the early lethality of the Lss mouse presumably associated with dehydration, it would be of interest to know if lethality of the mouse could be prevented by administration of fluids. Further, if this is the presumed cause of death and the identified patients clearly have epidermal loss of Lss, it’s unclear why their lifespans are extended in comparison to the mouse.

Reviewer #3: This manuscript by Wada et al. is a carefully crafted insight into LSS deficiency. The authors identified two sibling patients with hypotrichosis and other abnormalities, and performed exome sequencing leading to the identification of LSS mutations as the likely cause. Consistent with this, they found an increase in the substrate of LSS ((S)-2,2-epoxysqualene) in sebum from the patients – previous efforts had not been able to confirm a lack of LSS activity. They then made skin and lens-specific Lss-deficient mice to show that LSS deficiency results in similar skin problems and cataracts, as indicated in humans. This work is the first to confirm that a lack of LSS activity in the affected tissues leads to the specific phenotypes observed in patients with LSS deficiency. This is significant and has important implications for the treatment of this disease, as well as broader implications for other cholesterol synthesis enzyme defects associated with diseases. I have only minor comments, which I list below.

Minor comments:

1. Page 5, line 64: please rephrase “Cholesterol is an essential nutrient” as this is not strictly correct.

2. Supp Fig 1: Only LBR is listed – please also include DHCR14 (these enzymes catalyse the same step). Zymosterol is listed twice – it should be zymostenol in the Kandutsch-Russell pathway. Dehydrodesmosterol should be 7-dehydrodesmosterol. Dihydrolanosterol is, I think, more commonly referred to as 24,25-dihydrolanosterol.

3. Page 5, line 73-74: The sentence about the 24 patients is confusing – suggest rephrasing to “Out of 24 patients, all had either cataracts (5/24) or hypotrichosis (20/24), and one had both”.

4. Page 5, line 86: Should this be a citation to the publication by Dickinson et al 2016 Nature as indicated on the website?

5. Page 9, line 151-152: The sentence about W581R and G588S is confusing – consider revising or removing.

6. Fig 2c: The authors note that lanosterol was undetectable in the skin of these mice, but in humans, and the tamoxifen-inducible knockout mice, they were able to detect lanosterol and present a ratio. Could the authors speculate on why lanosterol was not detectable? Further, was there any difference in the levels of cholesterol itself?

7. Fig 2b, S9 Fig: Can these be presented with individual data points instead, as in other graphs?

8. Fig 2d: Was the decrease in body weight of Lssf/f-K14 mice significantly different from the others? This should be indicated in the text/figure.

9. The figure legends include some results which may be better placed in the main text – please consider revising.

10. Fig 4c: The authors indicate that lanosterol levels are not significantly different – could the authors comment on why they are unchanged? Are total cholesterol levels changed?

11. S7 Fig: What is the large peak? Is it cholesterol?

12. S12 Fig: What happens if control mice are treated with a topical application of 4-hydroxytamoxifen?

13. Some of the supplementary material could be included in the main figures e.g. S5, S7, S9 – I will leave this for the authors to consider.

Typographical/visual comments:

1. Page 5, line 66: cholesterol and intermediates are “precursors” to vitamin D, etc, rather than “components” of. Please rephrase.

2. Page 8, line 135: The mutation is mislabelled as Glu – should be Gln.

3. Page 13, line 226: Should this be an increased ratio rather than decreased?

4. In some places, LSS is referred to as lanosterol synthase, but in others it is lanosterol synthetase – please be consistent.

5. Page 8, line 145: Please define EB-LCLs (and PBMCs page 18, line 342 and fig 1 legend).

6. Fig 3b: Please label what the top and bottom pictures represent.

7. Fig 4a,b,c: For clarity, please consider labelling mouse/lens/H&E/immunohistochemistry.

8. S4 Fig: Please indicate which patient.

9. Fig 1f: Please align labels over bands.

**Have all data underlying the figures and results presented in the manuscript been provided?**

Reviewer #1: Yes

Reviewer #2: Yes

Reviewer #3: Yes

PLOS authors have the option to publish the peer review history of their article (what does this mean?). If published, this will include your full peer review and any attached files.

Reviewer #1: No

Reviewer #2: No

Reviewer #3: Yes: Laura Sharpe

---

## [Decision Letter · Decision Letter 1]

29 Dec 2019

Dear Dr Kikuchi,

Thank you very much for submitting your Research Article entitled 'Metabolic and pathologic profiles of human LSS deficiency recapitulated in mice' to PLOS Genetics. Your manuscript was fully evaluated at the editorial level and by independent peer reviewers. The reviewers appreciated the attention to an important topic but identified some aspects of the manuscript that should be improved.

We therefore ask you to modify the manuscript according to the review recommendations before we can consider your manuscript for acceptance. Your revisions should address the specific points made by each reviewer.

[LINK]

Yours sincerely,

Luke Engelking

Guest Editor

PLOS Genetics

Gregory Barsh

Editor-in-Chief

PLOS Genetics

Editor comments to author:

Thank you for your revised manuscript, which is now acceptable for publication pending the clarification of the textual description of Fig. S13, as suggested by reviewer 2.

Reviewer's Responses to Questions

**Comments to the Authors:**

Reviewer #1: The authors have extensively addressed all reviewers' concerns and have added new data to support their findings. Overall, this study is an interesting one; and the data presented here will act as a springboard for further investigations, particularly mechanistic ones, into disorders due to cholesterol biosynthesis defects.

Reviewer #2: In the manuscript entitled "Metabolic and pathologic profiles of human LSS deficiency recapitulated in mice ", the authors demonstrate disruption of lanosterol synthetase in an inducible mouse model of LSS deficiency results in biochemical deficits and tissue specific changes present within affected human subjects. The work presented within the manuscript is important to the fields of cholesterol biosynthesis defects, rare disease research, and epidermal function. The authors nicely demonstrate the identification of defined mutations within a small patient cohort of LSS deficiency and recapitulate some of the tissue specific deficits within a newly derived Lss epidermal-specific knockout mouse.

From a positive perspective, the authors attempted to answer the issues raised by reviewers after the initial submission. The manuscript is improved from the first submission and the authors have added some additional supplementary data, attempted to answer biochemical and signaling questions posed by reviewers, and clarified the text. However, as noted in my initial review, I still believe the work as presented is lacking a significant advancement forward for LSS deficiency. Though the study has demonstrated loss of LSS function within the epidermis results in neonatal lethality and lens specific Lss knockout induces tissue phenotypes, there is still a lack of data demonstrating the mechanisms downstream of Lss loss leading to lethality or tissue phenotypes.

- Biochemically, there appears to be no loss of epidermal cholesterol (Fig S9) which would mechanistically explain the barrier deficiency presented in Figure 2. It’s possible 2,3-epoxysqualene accumulation at picomolar/mg levels as demonstrated in Figure 2 could exert oxidative stress on the epidermis, inducing apoptosis or lipid peroxidation. However, this is not explored. While I am distinctly aware of the difficulty with these types of analyses in limited tissue samples, these are critical data.

- While the authors did test LSS effects on SHH and Wnt signaling in MEF isolations, no deficits were identified. It’s not clear if this lack of effect is due to a true maintenance of signaling in the absence of LSS activity or limitations due to the assay. Possible explanations that come to mind include cell type of choice representing mesodermal vs ectodermal lineage of the epidermis, method of LSS deletion via AAV1 (though the qPCR analysis demonstrates fairly efficient knockdown of Lss), or limitations of the qPCR assay used. The methodology suggests MEFs were isolated with FBS, AAV1 transduction performed, and RNA isolated 3 days post-transduction. Was the biochemical defect confirmed at this time point? Further, since both SHH and Wnt signaling utilize morphogen gradients through ligand secretion by one cell type and generally ligand-responsive signaling in a separate cell, I believe IHC analyses specific to the epidermis for SHH and/or Wnt responsiveness would be a better assay of choice. However, based on a lack of cholesterol loss within the epidermis (Fig S9), it's not clear if either of these pathways would be substantially disrupted in this tissue.

In my view, any of these three outcomes would represent a significant advancement for Lss deficiency:

1) identification of causative pathways downstream of LSS loss

2) determining the biochemical changes responsible for the pathological findings

3) correction (or prevention) of pathological findings.

Any of these would be acceptable. But as the work currently stands and in comparison to previously published studies on LSS, the major finding of epidermal or lens-specific loss of Lss with confusing biochemistry (no cholesterol loss) and no well-defined mechanism is troublesome.

Minor issues to clarify related to newly presented data

S13 - “There were fewer retinal ganglion cells in Lssf/f-Pax6 mice than there were in Lssf/f mice. In contrast, the outer nuclear layers were not found to be different between Lssf/f-Pax6 and Lssf/f mice.”

” – this is not actually quantified or explained well. What does ‘different’ mean?

Reviewer #3: The authors have done their best to address comments by the reviewers. The measurement of additional sterols would indeed be interesting, but is quite difficult and so it is understandable that this has not been as extensive as hoped.

**Have all data underlying the figures and results presented in the manuscript been provided?**

Reviewer #1: Yes

Reviewer #2: Yes

Reviewer #3: Yes

PLOS authors have the option to publish the peer review history of their article (what does this mean?). If published, this will include your full peer review and any attached files.

Reviewer #1: No

Reviewer #2: No

Reviewer #3: No

---

## [Editor Report · Decision Letter 2]

26 Jan 2020

Dear Dr Kikuchi,

We are pleased to inform you that your manuscript entitled "Metabolic and pathologic profiles of human LSS deficiency recapitulated in mice" has been editorially accepted for publication in PLOS Genetics. Congratulations!

Yours sincerely,

Luke Engelking

Guest Editor

PLOS Genetics

Gregory Barsh

Editor-in-Chief

PLOS Genetics

Comments from the reviewers (if applicable):

**Data Deposition**

http://datadryad.org/submit?journalID=pgenetics&manu=PGENETICS-D-19-01355R2

**Press Queries**

---

## [Editor Report · Acceptance letter]

21 Feb 2020

PGENETICS-D-19-01355R2 

Metabolic and pathologic profiles of human LSS deficiency recapitulated in mice 

Dear Dr Kikuchi, 

We are pleased to inform you that your manuscript entitled "Metabolic and pathologic profiles of human LSS deficiency recapitulated in mice" has been formally accepted for publication in PLOS Genetics! Your manuscript is now with our production department and you will be notified of the publication date in due course.

With kind regards,

Kaitlin Butler

PLOS Genetics

On behalf of:
